# Towards Understanding Therapeutic Failures in Masquelet Surgery: First Evidence that Defective Induced Membrane Properties are Associated with Clinical Failures

**DOI:** 10.3390/jcm9020450

**Published:** 2020-02-06

**Authors:** Marjorie Durand, Laure Barbier, Laurent Mathieu, Thomas Poyot, Thomas Demoures, Jean-Baptiste Souraud, Alain-Charles Masquelet, Jean-Marc Collombet

**Affiliations:** 1French Military Biomedical Research Institute (IRBA), BP73, 91220 Brétigny-sur-Orge, France; laure.barbier@intradef.gouv.fr (L.B.); thomas.poyot@intradef.gouv.fr (T.P.); collombet.irba@orange.fr (J.-M.C.); 2Percy Military Hospital, 101 avenue Henri Barbusse 92140 Clamart, France; laurent_tom2@yahoo.fr; 3Department of Surgery, French Medical Health Service Academy, Ecole du Val-de-Grâce, 1 place Alphonse Laveran 75005 Paris, France; 4Bégin Military Hospital, 69 avenue de Paris, 94160 Saint-Mandé, France; thomasdemoures@yahoo.fr (T.D.); jeanbaptiste.souraud@intradef.gouv.fr (J.-B.S.); 5Saint-Antoine Hospital, 184 rue du Faubourg Saint-Antoine, 75012 Paris, France; acmasquelet@free.fr

**Keywords:** induced membrane, Masquelet technique, bone defect repair

## Abstract

The two-stage Masquelet induced-membrane technique (IMT) consists of cement spacer-driven membrane induction followed by an autologous cancellous bone implantation in this membrane to promote large bone defect repairs. For the first time, this study aims at correlating IMT failures with physiological alterations of the induced membrane (IM) in patients. For this purpose, we compared various histological, immunohistochemical and gene expression parameters obtained from IM collected in patients categorized lately as successfully (Responders; *n* = 8) or unsuccessfully (Non-responders; *n* = 3) treated with the Masquelet technique (6 month clinical and radiologic post-surgery follow-up). While angiogenesis or macrophage distribution pattern remained unmodified in non-responder IM as compared to responder IM, we evidenced an absence of mesenchymal stem cells and reduced density of fibroblast-like cells in non-responder IM. Furthermore, non-responder IM exhibited altered extracellular matrix (ECM) remodeling parameters such as a lower expression ratio of metalloproteinase-9 (MMP-9)/tissue inhibitor of metalloproteinases (TIMP-1) mRNA as well as an important collagen overexpression as shown by picrosirius red staining. In summary, this study is the first to report evidence that IMT failure can be related to defective IM properties while underlining the importance of ECM remodeling parameters, particularly the MMP-9/TIMP-1 gene expression ratio, as early predictive biomarkers of the IMT outcome regardless of the type of bone, fracture or patient characteristics.

## 1. Introduction

Large bone defects usually originate from injuries resulting from either high velocity impact or heavy weight object crushing (e.g., gun shot, road accident) as well as bone loss caused by sepsis-triggered osteomyelitis, radiotherapy-induced osteonecrosis, or bone tumor resections [1]. Such bone injuries are defined as critical-sized bone defects, since they cannot spontaneously heal and they remain a challenge, even today, in reconstructive trauma surgery.

Different surgical techniques have been developed to address this challenge. The three most reported surgical techniques are the free vascularized fibular graft, the Ilizarov technique by local bone transportation and the Masquelet induced-membrane technique (IMT). The free vascularized fibular graft procedure requires excellent surgical experience due to the necessary microsurgical approach and it alters the donor healthy limb [2]. The Ilizarov technique is a slow and painful process and requires a great deal of expertise to perform successfully [3].

For its part, the Masquelet IMT is described as a technically simple, reliable, and reproducible procedure for which bone union time is independent of the bone defect size [4,5]. This two-stage approach is based on the induction of a foreign-body granulation membrane, called an “induced membrane” (IM), which favors bone regeneration. In practice, in the first stage, a polymethyl-methacrylate (PMMA) cement spacer is inserted between the bone defect ends following radical debridement and stabilization of the lesion. The spacer has a dual role since it prevents fibrous tissue invasion of the defect area and it induces formation of the surrounding membrane subsequent to a foreign body reaction (FBR) generated by the nature of the spacer itself. After 6 to 8 weeks, a second operative stage is performed. Here, the IM is cautiously incised to remove the cement spacer and an autologous bone graft is implanted into the biological chamber delimited by the membrane; bone repair then ensues for several months. In the literature, the IM is described as a pseudo-synovial tissue having biological properties. Indeed, the membrane secretes various growth factors [6,7,8,9] including transforming growth factor beta 1 (TGF-β1), fibroblast growth factor 2 (FGF-2), bone morphogenetic protein 2 (BMP-2) and vascular endothelial growth factor (VEGF), provides a well-vascularized environment and contains mesenchymal stem cells (MSC), the osteoprogenitors able to differentiate into mature osteoblasts, the cells responsible for bone formation activity [10,11].

Even though the IMT has shown to be very effective for the reconstruction of large diaphyseal bone defects (up to 20 cm wide), the rate of unsuccessful bone repair ranges between 9% and 59%, with a failure mean rate of approximately 18% [12]. Some clues explaining the complications and failures in IMT are provided in the literature. Data from several sources highlight either the recurrence of sepsis, active smoking, or lack of rigid fixation after the second operative step as participants in IMT clinical failures [13,14,15,16]. However, available studies in the area are, to date, either prospective or retrospective epidemiologic studies. No research has yet investigated the biological reasons behind the IMT failure at the IM level. Our hypothesis was that defective IM properties could account for IMT clinical failure. To address this issue, IM samples from 11 different Masquelet surgeries were collected and analyzed by using histological staining, immunohistochemistry, and gene expression analysis. Then, at the end of the therapy, we compared the biological properties of the IM samples between patients achieving successful bone regeneration (Responder patients, *n* = 8) and patients that failed to show consolidation (Non-responder patients, *n* = 3). As expected, we discovered differences in IM properties between the groups, including histological alterations and changes in osteoprogenitor content. Most interestingly, we showed that MMP-9-dependent remodeling of the extracellular collagen matrix in non-responding patients was altered. In addition, we were able to introduce the MMP-9/TIMP-1 mRNA ratio as a putative biomarker to predict the osteogenic capacity of the IM to achieve successful bone repair during the second operative stage, regardless of patient characteristics or IMT failure associated-risk factors.

## 2. Experimental Section

### 2.1. Study Design

This study was conducted in accordance with the declaration of Helsinki and approved by the local ethical committee of the Percy Military Hospital (20.19PPRC.03). We randomly enrolled eleven patients who received surgical treatment of non-unions subsequent to long bone fractures in the Department of Orthopedics and Trauma Surgery of the Percy Military Hospital (Table 1). All participants gave informed consent according to institutional guidelines. We purposely chose a heterogeneous cohort rather than stratify it for risk factors because our assumption was that IMT failure could be related to defective IM properties regardless of risk factors and patient characteristics.

Induced membrane tissue was collected in sterile conditions during the routine surgical removal of the gentamycin-PMMA spacers (Palacos© R + G) in the second stage of the IMT. To ensure proper orientation of the membrane, tattoo ink was placed on the surface of the lining membrane that abutted the PMMA cement at the time of surgery.

Immediately after collection, the membranes were washed in phosphate-buffered saline solution (PBS) and divided up for the different treatments related to subsequent analysis: 4% paraformaldehyde was used for the histological analysis, RNAlater stabilization solution was used for the polymerase chain reaction analysis, and fresh DMEM cell culture media for the cell culture.

Enrollment in the study did not change the usual patient follow-up based on the clinical and radiological assessment of fracture healing at 6 weeks, 4 and 6 months postoperatively. Bony union was defined as three-sided cortical bridging on two perpendicular radiographic views. Clinically, healing was assessed by the stability of the fracture zone and the absence of pain and full weight bearing in the case of the lower limb. At the 6 month follow-up consultation, final outcome of the IMT was evaluated by two senior orthopedic surgeons and their teams. Patients were declared Responder (R) (successful healing, *n* = 8) or Non-Responder (NR) (unsuccessful healing, *n* = 3) to the IMT therapy. Inclusion of patients in either group was communicated by the surgeons to the research team at the end of the study, to control against bias.

### 2.2. Histology and Immunohistochemical Analysis

After 1X PBS wash, membrane fragments were fixed for 24 h in 4% paraformaldehyde. Fragments were then processed for paraffin histology by dehydrating through a graded alcohol series and cleared in xylene before embedding in paraffin wax. Five-micrometer-thick sections were cut from each block using a Leica Microtome (Leica MicroSystems GmbH, Wetzlar, Germany) and mounted onto silanized slides for histological and immunohistochemical examination. Routine hematoxylin eosin saffron (HES) staining was performed for tissue and cell identification. Collagen fibers of the stroma were evaluated by staining slices with picrosirius red dye prior observation under polarized light using a Leica DM6000B microscope. Stroma reticular fibers were stained by reticulin kit (RE-100T, Biognost, Zagreb, Croatia) and observed under a Leica DM2000.

Concerning immunohistochemical analysis, deparaffinized sections were first treated with 3% hydrogen peroxide to block endogenous peroxidase activity. Then, sections were labeled with anti-CD68 (clone KP1, Roche Diagnostics, Rotkreuz, Switzerland), anti-CD31 (clone JC70, Roche Diagnostics, Rotkreuz, Switzerland) or anti-matrix metalloproteinase 9 (MMP-9, (#58803, Abcam, Cambridge, UK) primary antibodies to detect macrophages and myeloid elements (CD68), endothelial cells from mature blood vessels (CD31) or gelatinase involved in the extra cellular matrix remodeling (MMP-9). After washing, visualization of CD68- and CD31-stained sections was performed using an HRP-conjugated secondary antibody followed by colorimetric detection using the Ultraview DAB kit (Roche Diagnostics, Rotkreuz, Switzerland). Tissues were counterstained with hematoxylin. For MMP-9 immunostained sections, the ImmPREss system (#MP-7402, Vector Laboratories, Burlingame, CA, USA) was used as the secondary antibody reagent and counterstaining was achieved with Meyer’s hemalun.

### 2.3. Isolation and Characterization of Mesenchymal Stromal Cells (MSC) from IM Fragments

MSCs were isolated according to the explant culture method. Briefly, IM fragments (approximately 5 mm^3^ total volume) were transferred to culture dishes. Primary explant cultures were established in Modified Eagle’s medium (MEM) containing 5% human platelet lysate (provided by the CTSA, Clamart, France), heparin (2 U/mL) and ciprofloxacin (500 µg/mL). MSCs were incubated at 37 °C with 5% CO_2_ and 95% humidity for 3 weeks. At 80–100% confluence, cells were trypsinized and MSC purity was assessed by flow cytometry using the following monoclonal antibodies: anti-CD45 clone HI30 Peridinin-chlorophyll protein cyanine (PerCP-Cy™5.5), anti-CD90 clone 5E10 fluorescein isothiocyanate (FITC), anti-CD73 clone AD2 allophycocyanin (APC), anti-CD105 clone 266 phycoerythrin (PE) (BD Pharmingen). Data acquisition and analysis were performed with LSR II flow cytometer and FlowJo software, respectively (BD Biosciences).

### 2.4. Real-Time PCR Analysis

Membrane tissues intended for molecular biology analysis were collected and stored in RNAlater^®^ (Ambion, USA). Samples were kept at 4 °C for 24 h and then stored at −20 °C before homogenization in guanidium-based lysis buffer using a TissueLyser II (RLT buffer, Qiagen, 30 Hz, 3 X 2 min, two 3 mm-carbide beads). Total RNA was extracted using the RNeasy Fibrous Tissue mini kit (Qiagen, France) according to the manufacturer’s recommendations with an additional DNase step (Qiagen, France). RNA extracts were eluted with 30 µL of RNase free water. Each sample was checked for absence of contamination by measuring the optical density (230, 260 and 280 nm) using a microvolume spectrophotometer (Nanodrop 1000, Thermofisher scientific, USA). Reverse transcription was performed to synthesize cDNA from 400 ng total RNA using EuroScript reverse transcriptase according to the manufacturer’s instructions (Eurogentec #RT-RTCK-03). RNA integrity and RT yields were confirmed using the 5′/3′ integrity assay [17] with rplp0 selected primers. Primer design and optimization to prevent dimerization, self-priming and melting temperature were carried out using the MacVector^®^ software (Accelerys, USA). Primers were selected in intron flanking regions to exclude genomic DNA amplification and were assessed for human specificity using the nucleotide Blast algorithm. Oligonucleotide primers were synthesized by Eurogentec (Sereing, Belgium). Real-time qPCR was carried out using a LightCycler^®^ 480 (Roche Applied Science) with SybrGreen I Mastermix (Roche Applied Science, Mannheim, Germany). qPCR primers for monocyte chemoattractant protein-1 (MCP-1), CD68, runt-related transcription factor 2 (Runx2), Von Willebrand factor (vWF), transforming growth factor beta-2 (TGF-β2), matrix metalloproteinase-9 (MMP-9), and tissue inhibitor of metalloproteinase-1 (TIMP-1) are listed in Appendix A, along with concentrations and annealing temperatures optimized for each primer. mRNA expression quantification was measured using the comparative threshold method [18] with efficiency correction estimated from a standard curve. Normalization was assessed using GENORM software. Briefly, a geometric average of three internally validated reference genes (rplp0, ppia and ywhaz) was conducted [19]. The 0.12 pairwise variations of these three genes were below the threshold (0.15) that requires inclusion of an additional normalization gene.

### 2.5. Statistics

Statistical analysis was performed using GraphPad’s Prism 7 statistical software. Data were analyzed by the Shapiro–Wilk test to assess normal distribution. When the data sets met both the test requirements for distribution and variance, comparison of multiple groups was performed by using a one-way ANOVA with the Tukey’s post-hoc test. When the data did not meet one of the test requirements, a non-parametric Kruskal–Wallis test was performed; *p* < 0.05 was considered statistically significant. Data are expressed as the mean +/− standard error of the mean (SEM) for both responder and non-responder groups. For the MMP-9/TIMP-1 gene expression ratio, the Cohen’s d effect size, which reflects the magnitude of the observed difference, was calculated by using Statistica software. A calculated effect size of 0.8 was considered as “large” according to Cohen [20].

## 3. Results

### 3.1. Demographic and Clinical Features of Patients

As shown in Table 1, IM samples were collected from eleven surgeries. Eight out of eleven (72.2%) surgeries were responders (R) to the Masquelet therapy and the three others (27.2%) were non-responders (NR). Of note, patient NR2 and R4 is the same patient, undergoing two different and independent Masquelet surgeries over a one-year period to treat a femoral bone defect. Only three patients were treated for upper limb fracture (27.2%), and most of the patients had bone infections at the time of PMMA spacer placement (72.2%, 8 out of 11 fractures). There was no difference in the patient age and the bone defect length between both groups. However, membranes from NR patients displayed a significantly longer maturation time (23.00 +/−9.07 weeks vs. 10.00 +/− 1.55 weeks for NR and R groups respectively, *p* = 0.047).

### 3.2. Histology and Cellularity are Disrupted in Non-Responder Induced Membranes (IM)

Upon histological examinations (Figure 1), typical responder IM displayed a dense cell layer at the cement interface that included fibroblast-like cells and macrophages, below which was a radially outward layer including fibroblasts and blood vessels (Figure 1A). As for NR membranes (Figure 1B), the most obvious change concerned the lining layer at the PMMA interface, which was thinner (NR1, NR3) or absent (NR2), resulting in a striking decrease in cellularity. Indeed, quantification of cell nuclei within the membranes revealed that cell density was more than halved in NR membranes compared to the R group (802.1 +/− 115.6 cells/mm^2^ for NR group vs 1865.7 +/− 218.6 for R group, *p* = 0.012).

### 3.3. Distribution and Number of CD31-Positive Blood Vessels are not Altered in Non-Responder Induced Membranes

Since histological differences in NR membranes suggested changes in IM activity, we then focused on blood vessels, given that blood vessels are considered a cornerstone of the IM mechanism of action [21,22,23]. As seen in Figure 2A, CD31-labeled mature blood vessels were found throughout the membranes in both R and NR samples, specifically in the deep layer in contact with the muscle and immediately beneath the PMMA-adjacent layer. Quantification analysis (Figure 2B) did not reveal any difference in the number of CD31 positive blood vessels in NR membranes when compared to R membranes.

### 3.4. Mesenchymal Stromal Cell Content is Altered in Non-Responder Membranes

Besides providing vascular supply, the IM is also known to gather mesenchymal stromal cells (MSC) with osteogenic potential [10,24], thereby leading us to compare the MSC content between R and NR membranes (Table 2). Concerning R patients, seven out of seven (100%) tested IM samples were able to generate adherent cell colonies. After culture expansion, these IM-derived cells were characterized based on their surface antigen expression profile. Flow cytometric analysis showed that ex vivo-expanded cells displayed the MSC phenotypic marker CD45^-^ CD90^+^ CD73^+^ CD105^+^. Conversely, two out of two (100%) tested IM samples from NR patients did not generate any MSC expansion, supporting the view that the osteoprogenitor content of these membranes was altered.

### 3.5. CD68 Pan Macrophage Marker Localization and Gene Expression are Similar in both Responder and Non-Responder Induced Membranes

Since macrophages are known to play a decisive role in both the regulation of wound healing and the development of the foreign body reaction [25], we examined macrophage distribution in IM; suspecting that histological observations in NR membranes could be linked to changes in the macrophage distribution pattern. We visualized macrophages in the membranes by immunolabelling the pan monocyte/macrophage marker CD68. Surprisingly, no difference in CD68 staining was found between R and NR patients, as can be seen in their representative IM pictures (Figure 3). In both groups, CD68 labeling was found throughout the IM, and especially in cells localized adjacent to the PMMA spacer (Figure 3, boxed regions). The percentage of CD68-positive area in NR membranes did not differ significantly from that seen in the R membranes (7.13 +/− 1.86% for NR group vs 9.90 +/− 2.8% for R group, *p* = 0.611). Moreover, CD68 gene analysis confirmed that there was no difference in the expression of this pan-macrophage marker between R and NR membranes (*p* = 0.862, data not shown). Likewise, the expression of monocyte chemoattractant protein-1 (MCP-1) gene, coding a potent chemokine that regulates migration and infiltration of monocytes and macrophages, was unchanged between R and NR groups (*p* = 0.390, data not shown). Taken together, these data suggest that tissular inflammatory response and macrophage infiltration remained unchanged in NR patients.

### 3.6. MMP-9 Expression is Disrupted in Non-Responder Induced Membranes

To gain more insight into the physiological changes in NR membranes, we performed reverse transcription quantitative PCR analysis of RNA isolated from IM to examine the expression of some key bone-, vasculature- and remodeling-related genes. Expression levels of Runx2 mRNA (early osteogenic marker) and vWF mRNA (immature blood vessel marker and pro-angiogenic factor) were slightly decreased in NR membranes, though not significantly (*p* = 0.368 and *p* = 0.207, respectively, data not shown). Conversely, mRNA expression of transforming growth factor beta-2 (TGF-β2), which is involved in collagen synthesis, was increased in NR membranes, though also not significantly (*p* = 0.136, data not shown). Nevertheless, statistical significance was reached when considering the matrix metalloproteinase-9 (MMP-9) and tissue inhibitor of metalloproteinase-1 (TIMP-1) mRNA ratio. Indeed, as shown in Figure 4A, the MMP-9/TIMP-1 balance was 87% lower in NR membranes than in R membranes (*p* = 0.04). In agreement with this, the measure of the effect size by the Cohen’s d effect size was found equal to 3.4, meaning that the magnitude of the observed effect was very large. As the decreased ratio was mainly attributed to a reduced mRNA level of MMP-9, we performed immunohistochemical analysis to investigate MMP-9 protein expression. In line with the mRNA analysis, immunochemistry revealed that MMP-9 staining was barely detectable in NR samples, whereas it was widely expressed in R membranes (Figure 4B).

### 3.7. Collagen Matrix Organization is Disrupted in Non-Responder Induced Membranes

Based on the above features, we set out to explore the fibrous components of the IM extracellular matrix. Collagen content was analyzed using picrosirius red-stained membranes under polarized light (Figure 5A). As expected, NR membranes showed thicker and tightly packed fibers with reddish birefringence than R membranes. Moreover, reticulin staining revealed the presence of some thin reticular fibers around blood vessels in all NR membranes (three out of three membranes) whereas only two R membranes (two out of eight) displayed a few, sparse reticular fibers around blood vessels (Figure 5B). The six other R membranes were free of reticular fibers. Interestingly, orcein-stained elastic fibers were detected in NR membranes while they were absent in R membranes (Figure 5C).

Taken together, our data point out an alteration of the extracellular matrix remodeling turnover in NR membranes.

## 4. Discussion

IMT is considered as the method of choice to address posttraumatic or infected segmental bone loss [13,26]. Regardless of the heterogeneity and features of study populations (age, sex, smoking…) and fractures (osteosynthetic materials, causes of segmental defects…), numerous studies agreed that the IM is the key element of the surgery as it creates a biologically privileged-environment for bone healing [4,10,22,24]. Despite its importance, of the 18% of procedures that fail to achieve proper bone healing [12], no attention has been paid to the role of IMs in these failures. Thus, this preliminary study aimed at comparing the physiological properties of IMs between three fractures that failed to show bone consolidation (non-responder patients) and eight fractures treated successfully (responder patients). Given our assumption that IMT failure could be related to defective IM properties regardless of risk factors and patient characteristics, we purposely chose to use a heterogeneous cohort without patient stratifications.

As expected, substantial differences were observed between R and NR membranes. Upon histological examination, we found a 57% decrease in the cell nuclei number in NR membranes. This was mainly attributed to a reduced size of the PMMA adjacent-lining layer of the NR membranes, including a reduction in fibroblast-like cells. Moreover, our findings suggested an impaired osteoprogenitor content in NR membranes. As fibroblasts and MSC are an important source of cytokine secretion, to include growth factors and angiogenic factors [27], the massive cell density reduction observed in NR membranes will likely cause an impaired biological chamber that is partially deprived of cytokine secretions surrounding the bone autograft. To test this assumption, it would be interesting to perform further mass spectrometry-based analysis of IM secretion products.

Previously, studies have reported that macrophages are central cells in the initiation, duration, and outcome of the FBR [28,29,30]. Although macrophages are dispersed through FBR capsules, they mainly accumulate at the lining layer of the capsule along with fibroblasts at the interface with the implanted foreign body [31,32].

In NR membranes, we noticed that the cell layer in contact with the PMMA spacer was thinner (NR1, NR3) than in R membranes, or even absent (NR2), leading us to speculate about a disruption in monocyte/macrophage content in NR membranes. Therefore, we investigated the distribution pattern of the pan-macrophage marker CD68 [33]. Interestingly, no decrease in CD68 staining was found in NR membranes, confirmed by unchanged mRNA levels of CD68 and MCP-1. A likely explanation to this surprising result lies in the use of the CD68 marker. CD68 is commonly considered as a myeloid pan-macrophagic marker as it is highly expressed by monocytes, M1 and M2 macrophages, histiocytes, osteoclasts and foreign body giant cells. However, its expression is not unique to myeloid cells [33,34]. For example, it can also be found to a lesser extent on fibrocytes. Yu et al. recently indicated in a murine subcutaneous implantation model that macrophages dispersed throughout the FBR exhibited traits characteristic of the M2 anti-inflammatory phenotype while macrophages located at the interface displayed an M1 pro-inflammatory profile [32]. In our study, the lack of specificity of the CD68 marker makes it difficult to distinguish different cell types including macrophage subtypes. This can result in concealing potential changes to the balance of macrophage phenotypes in NR patients. Clearly, further targeted research on macrophages in the IM needs to be done to shed light on a potential myeloid component in IMT failure.

In the present study, we detected a significant lower expression ratio of MMP-9/TIMP-1 mRNA in NR membranes, which was mainly due to reduced MMP-9 gene expression. MMP-9, a zinc-dependent proteinase, is released by many cells, predominantly fibroblasts and inflammatory cells including neutrophils and macrophages [34]. Together with its endogenous inhibitor TIMP-1, it regulates numerous signaling pathways of pivotal importance in the wound healing process, inflammation, and FBR [35]. For example, previous experimental work reported that MMP-9 null mice displayed non-unions and delayed unions of their fractures. This was caused by persistent cartilage at the injury site due to the loss of MMP-9′s impact on periosteal stem cell fate and the distribution of inflammatory cells in the callus [36,37]. Regarding the FBR, MacLauchlan et al. [38] provided evidence that MMP-9 is required for macrophage fusion into foreign body giant cells and that subcutaneous foreign-body capsules in MMP-9 null mice are composed of mature and disrupted collagen fibers lacking organization. More recently, the importance of MMP-9 in the IMT context has been highlighted by Haubruck and colleagues [39]. Indeed, they reported that serum level of MMP-9 was higher in Responder patients than Non-Responders during the course of the IMT treatment. Our results confirmed and supplemented the Haubruck findings by revealing a decreased expression of MMP-9 in the IM.

The MMP-9/TIMP-1 balance is in part responsible for ECM turnover control. Thus, a shift in balance in favor of MMP-9 results in increased ECM proteolysis, whereas a shift in balance in favor of TIMP-1 results in protection of the ECM and decreased proteolysis [40]. In NR membranes, change in the MMP-9/TIMP-1 ratio was mainly attributed to a reduced MMP-9 gene expression while TIMP-1 mRNA level was only slightly increased; implying that NR membranes displayed a lower ECM turnover than R membranes. Consistent with this hypothesis, we observed a wider network of collagen fibers in NR membranes as compared to R membranes. A question which arises from this observation is whether the above-mentioned absence of IM-derived MSC in the NR group is a result of impaired cell migration out of the membranes due to the thick collagen network or merely an altered MSC content. Nevertheless, taken together our findings are the first ones to highlight the IM contribution to IMT failure. We showed that IMT failure was associated with IM extracellular matrix alterations, regardless of patient and fracture features. Furthermore, considering the slightly increased mRNA expression of the profibrotic factor TGF-β2 [41] in the NR group, our data support the postulation that NR membranes display some pro-fibrotic features. Further research would be needed to substantiate this assertion.

The use of a small and heterogeneous cohort can be considered the main limitation of our study. Indeed, findings presented here were based upon a small cohort of membranes with both short and long maturation times (6 to 40 weeks old) and derived from either upper or lower limb fractures that were stabilized by various types of fixation. A recent work from Gindraux et al. [42] suggested that IMs maintained their osteogenic properties even when the second stage of the surgery is delayed. Besides Gindraux’s report, the impact of the risk factors on the IM properties are wholly underexplored, even though risk factors are likely to influence important properties of the IMs, such as vascularization. Small size and heterogeneity of the cohort might explain, at least in part, why no difference in the number of CD31 positive blood vessels was detected in NR membranes when compared to R membranes. On the other hand, the small and heterogeneous cohort is also a strength of our study when considering the finding of the MMP-9/TIMP-1 mRNA ratio. Indeed, the magnitude of the difference between R and NR groups was so large, as substantiated by the calculated Cohen’s d effect size of 3.4, that we were able to detect a significant effect in such a small and heterogeneous cohort. This strengthens the robustness of the MMP-9/TIMP-1 mRNA ratio as a putative biomarker of IMT failure. Further studies using larger and heterogeneous cohorts are warranted to verify this outstanding finding.

In conclusion, the present study was the first to highlight such IM physiological changes related to IMT failure, independently of the type of bone or fixation, IM maturation time or other patient characteristics. This IMT failure was associated with defective IM properties including decreased cellularity, decreased osteoprogenitor content and impaired ECM remodeling. In addition, we introduced the MMP-9/TIMP-1 mRNA ratio as a potential biomarker to predict IMT final outcome.

## Figures and Tables

**Figure 1 jcm-09-00450-f001:**
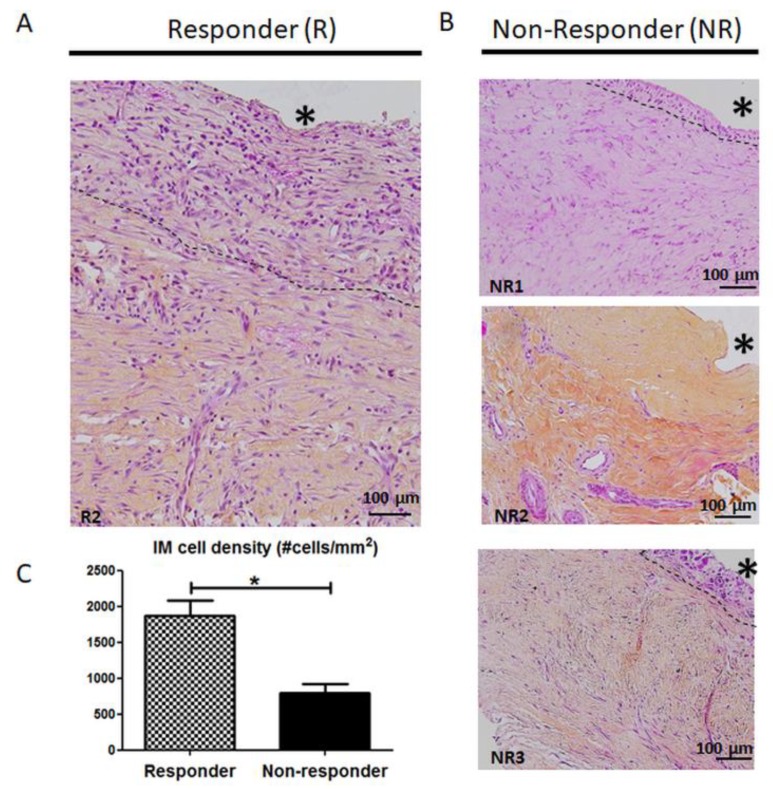
Representative hematoxylin and eosin-stained IM for (**A**) responder patient (patient R2) and (**B**) non-responder patients (NR1, NR2 and NR3). Responder patients present well organized IM characterized by a rich cellular network in contact with polymethyl-methacrylate (PMMA, * indicates the PMMA site before its removal) and a thick vascularized layer in contact with muscles. In comparison, non-responder patients display poorly structured and poorly cellularized IM (scale bar = 100 μm). (**C**) Cell density in IM from responder and non-responder patients. Results are shown as the mean +/- SEM, * *p* < 0,05.

**Figure 2 jcm-09-00450-f002:**
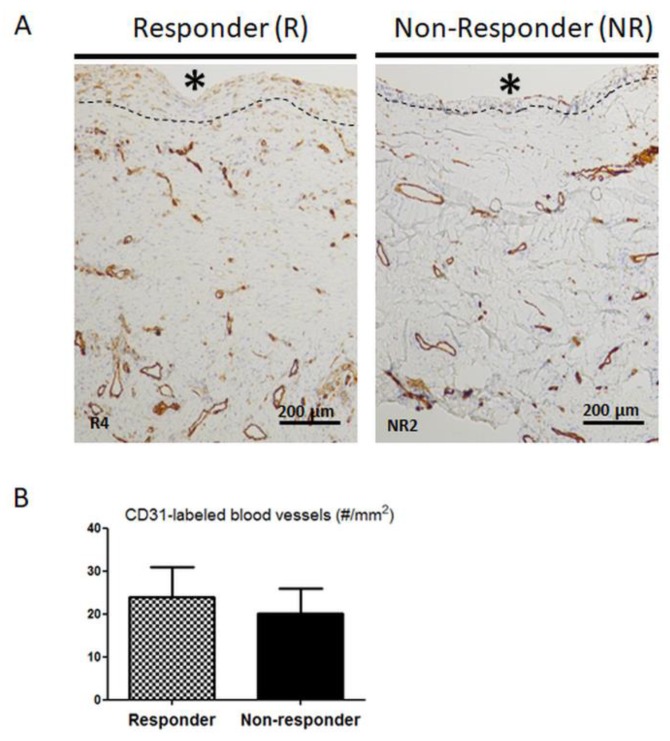
(**A**) CD31-positive mature blood vessels are found throughout both responder (R) and non-responder (NR) membranes, but predominantly in the deep layer in contact with the muscle and immediately beneath the PMMA-adjacent layer (dashed line). * indicates the PMMA site before its removal, scale bar = 200µm. (**B**) The number of CD31-labeled blood vessels was counted in six non overlapping microscopic fields of the induced membranes. Results are shown as the mean +/− SEM.

**Figure 3 jcm-09-00450-f003:**
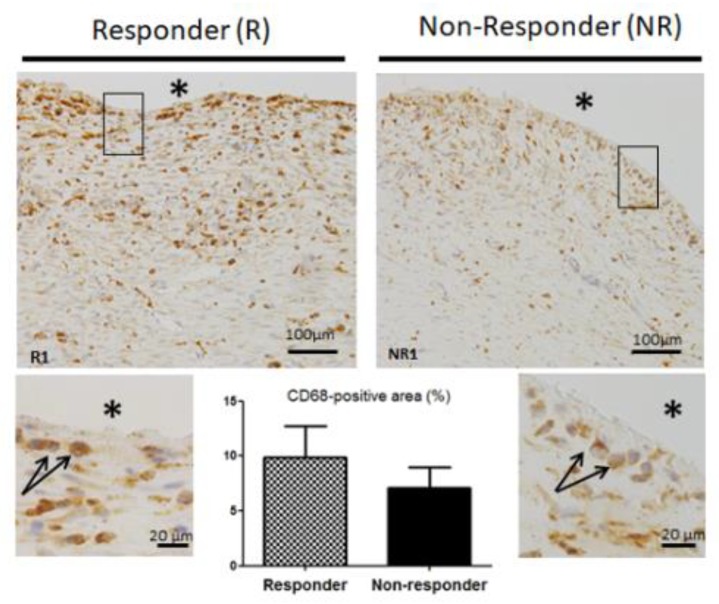
Representative images of CD68 immunohistochemistry. Black star (*) indicates PMMA location before its removal (scale bars 100 μm). Boxed regions are shown at higher magnification below (scale bars 20 μm). Black arrows point to CD68-positive cells. Quantitative analysis of the percentage of CD68-positive area was made in three non-overlapping microscopic fields of the induced membranes. Data are expressed as the mean ± SEM.

**Figure 4 jcm-09-00450-f004:**
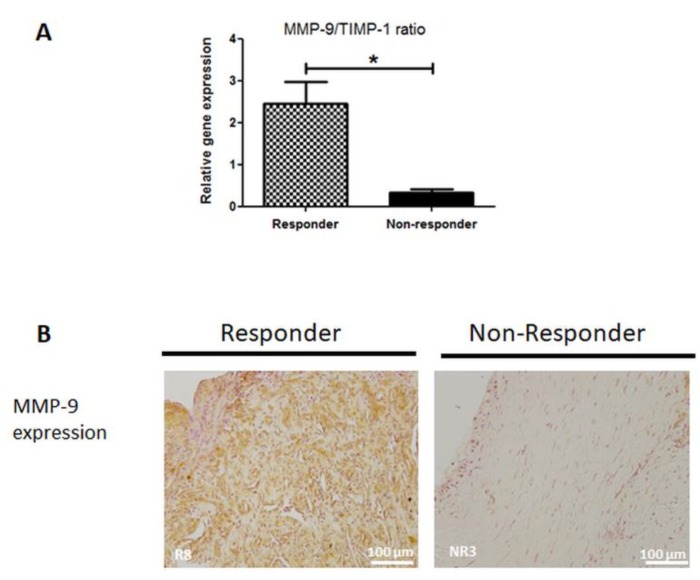
(**A**) Relative quantification of metalloproteinase-9 (MMP-9)/tissue inhibitor of metalloproteinases (TIMP-1) ratio mRNA in responder (R) and non-responder (NR) induced membranes. Data are expressed as the mean ± SEM, * *p* < 0.05. (**B**) MMP-9 immunolocalization in R and NR induced membranes. Scale bars = 100 µm.

**Figure 5 jcm-09-00450-f005:**
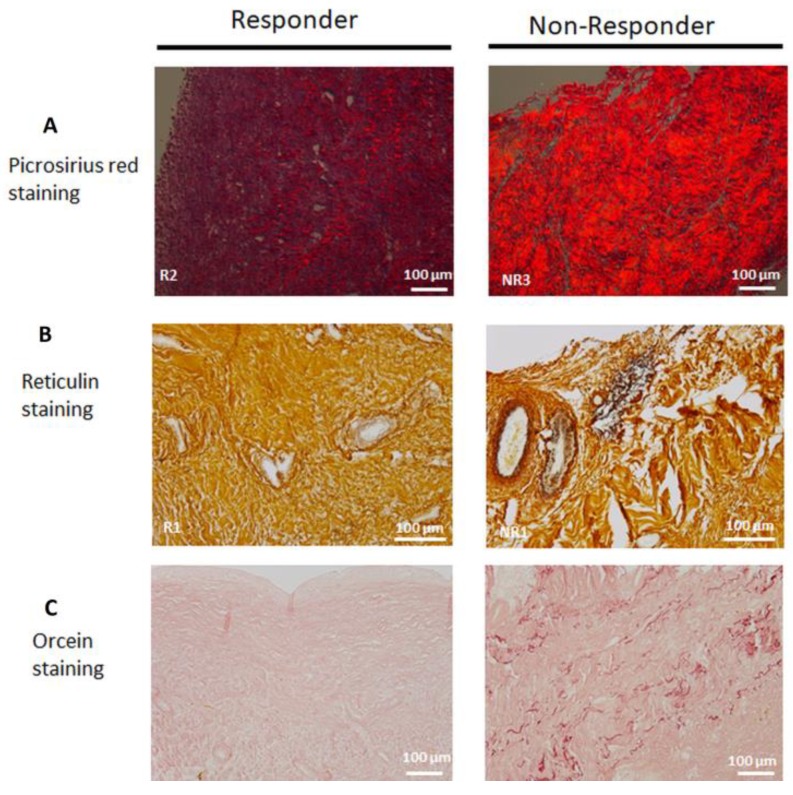
Representative pictures of picrosirius red (**A**), reticulin (**B**) and orcein (**C**) staining in responder and non-responder induced membranes. Scale bars = 100 µm.

**Table 1 jcm-09-00450-t001:** Demographic and clinical characteristics of patients.

	Patient Identification	Age (yrs)	Gender (F/M)	Smoker (Y/N)	Bone Defect Location	Bone Defect Lenght (cm)	Fracture Fixation	Infection (Y/N)	Duration Spacer in Site (wks)
Responder (R)	R1	30	M	N	Left tibia	12	Ex.Fix	N	20
	R2	65	M	N	Left radius	7	Plate	Y	12
	R3	87	M	N	Left tibia	5	Nail	Y	6
	R4	40	M	N	Left femur	9	Nail	N	8
	R5	46	M	Y	Left tibia	8	K-wires	Y	8
	R6	55	M	N	Left tibia	4	Ex.Fix	Y	10
	R7	42	M	N	Right tibia	15	Ex. Fix	Y	8
	R8	40	M	Y	Right radius	6	Plate	Y	8
Non responder (NR)	NR1	47	F	N	Left tibia	10	Ex. Fix	Y	20
	NR2	40	M	N	Left femur	9	Nail	N	40
	NR3	60	M	Y	Right humerus	7	Plate	Y	9
Mean +/− SEM	Responder	50.63 ± 6.40				8.25 ± 1.30			10.00 ± 1.55
	Non-responder	49.00 ± 5.85				8.66 ± 0.88			23.00 ± 9.07
*P*-value		0.88				0.85			0.047

R = responder; NR = non-responder; F = female; M = male; Y = yes; N = no; Ex.Fix = external fixator. * Refers to the same patient undergoing two different and independent Masquelet surgeries.

**Table 2 jcm-09-00450-t002:** Presence or absence of ex-vivo expanded mesenchymal stromal cells in responder (R) and non-responder (NR) patients. IM-derived cells were cultured and characterized as mesenchymal stromal cells (MSC) based on their surface marker expression.

	Patient Identification	Presence or Absence of IM-Derived MSC (CD45^-^ CD90^+^ CD73^+^ CD105^+^ Cells)
Responder (R)	R1	Presence
	R2	Not tested
	R3	Presence
	R4	Presence
	R5	Presence
	R6	Presence
	R7	Presence
	R8	Presence
Non-Responder (NR)	NR1	Not tested
	NR2	Absence
	NR3	Absence

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
