# Peer review of "Towards Understanding Therapeutic Failures in Masquelet Surgery: First Evidence that Defective Induced Membrane Properties are Associated with Clinical Failures"

_jcm, 2020, doi:10.3390/jcm9020450_

Round 1

Reviewer 1 Report

Dear authors,

thank you for your elaborate response to the concerns I had regarding your first submitted draft. I fuly acknowledge the thought and plannning that have gone into this endeavour you have undergone. As a trauma surgeon having utilized IMT over 200 times in the last 9 years, I am somewhat astonished that it took 3 years to compile the 11 patients you have reported on. It is also difficult for me to accept this as your main argument supporting your numbers. And it remains, in my point of view, wholly inacceptable to draw the conclusions you have made from this small a sample size. The properties you have described leading to failure of IMT may be completely arbitrary and also occur (albeit somewhat less) in cases who have successfully healed - with your numbers, you will never know. As the mechanisms by which risk factors influence IMT are not well-examined in regards of their pathology, I find it highly speculative that you conclude that there is a causal connection between the two.

In regards to point 3, while I accept that the basic principles of bone healing are the same in all anatomical regions, there are marked differences in the trauma mechanisms and the way they influence the five aspects of the diamond concept, especially if infected and non-infected cases are mixed. I therefore must contradict your assumption about IMT failure- it is most definitely NOT solely a question of the properties of the IM.

Concerning Cohen's d, in my recollection this is used for calculating sample sizes in advance and not for retrospectively justifying a low sample size with an effect size which might have been obtained by chance. I will follow this further with one of our statisticians as I find it interesting.

In regards to point 5, Gindraux's PRELIMINARY study was based on 7 patients out of 34 for whom testing was split into 2 groups of 4 and 3 patients respectively. This is a weak argument considering this publication:

J Tissue Eng Regen Med. 2016 Oct;10(10):E382-E396. doi: 10.1002/term.1826. Epub 2013 Nov 8.
Establishment and characterization of the Masquelet induced membrane technique in a rat femur critical-sized defect model.
Henrich D1, Seebach C2, Nau C2, Basan S2, Relja B2, Wilhelm K2, Schaible A2, Frank J2, Barker J3, Marzi I2.

Which assumes the contrary, albeit in a rat model.

Regarding Point 6 you have not proven or given literature to prove that while smoking has been shown in numerous studies to obviously influence clinical results of bone defect therapy, you do not know if it affects the biological basics of membrane properties and therefore decide to discard this fact. I find that quite astounding.

Points 7 and 8 are answered in an analogous manner to the previous points, so I will refrain from repeating myself.

Point 9: thank you for clarifying this.

Point 10: I deduce from your response that the operating surgeons evaluated their own cases radiologically and clinically. Do you not consider this a source of potential bias?

Point 11: Again: what proof do you have that biomarker levels, while admittedly being ubiquitous, are truly independent of the specific parameters of the clinical case? Do you have sources to support this assumption?

Thank you very much for your comments, work and time.

Reviewer 2 Report

The paper is suitable for publication in the present form.

This manuscript is a resubmission of an earlier submission. The following is a list of the peer review reports and author responses from that submission.

Round 1

Reviewer 1 Report

Thank you for the chance to review your paper.

This study, while presenting an interesting approach to deeper understanding of the mechanisms behind bone regeneration via masqulet-technique and epecially its limits, has several methodological issues that need clarification:

Why did you limit yourself to merely 11 cases?

Why were there only 3 non-responders included?

Why did you not limit yourself to one anatomical region (i.e. lower extremity)?

Why did you not perform a power analysis of your statistics?

How can you be sure that wildly differing gestation periods of the IM do not affect your results?

Why did you not exclude smokers from your cohort, as this is a well-known risk factor that is bound to corrupt your results at such a low number? Why did you not discuss this?

Why did you not further stratify your cohort for other risk-factors?

Why do you deem it acceptable to mix bot infected and non-infected non-unions in your cohort? Do you know how this affects your histological and lab results?

How did you evaluate response vs. non-response? Via X-ray or CT? Were clinical results (weight-bearing, etc.) considered?

How many raters evaluated R vs. NR? Were these the surgeons themselves?

How can you be sure that the results you are discussing are not confounded by the abovementioned limitations which your study has and how can you be sure that the hetereogenities you mention in lines 378 ff. had no effect on your observations?

Please elaborate.

Thank you.

Reviewer 2 Report

The concerns regards, as stated by the authors as well, the small cohort, the eterogeneity of treatment and fracture site, the upper or lower limb distribution, the maturation times etc. but overall the study is brilliant and deserves credit. I hope it might offer further impulse for shedding new light of the mechanism of cure and its quantification through biomarkers studies such as MMP9/TIMP ratio.